# Adaptive Active Hypothesis Testing under Limited Information

**Fabio Cecchi**
Eindhoven University of Technology, Eindhoven, The Netherlands
f.cecchi@tue.nl

**Nidhi Hegde**
Nokia Bell Labs, Paris-Saclay, France
nidhi.hegde@nokia-bell-labs.com

## Abstract

We consider the problem of active sequential hypothesis testing where a Bayesian decision maker must infer the true hypothesis from a set of hypotheses. The decision maker may choose for a set of actions, where the outcome of an action is corrupted by independent noise. In this paper we consider a special case where the decision maker has limited knowledge about the distribution of observations for each action, in that only a binary value is observed. Our objective is to infer the true hypothesis with low error, while minimizing the number of action sampled. Our main results include the derivation of a lower bound on sample size for our system under limited knowledge and the design of an active learning policy that matches this lower bound and outperforms similar known algorithms.

## 1   Introduction

We consider the problem of active sequential hypothesis testing with incomplete information. The original problem, first studied by Chernoff [1], is one where a Bayesian decision maker must infer the correct hypothesis from a set of $J$ hypotheses. At each step the decision maker may choose from $W$ actions where the outcome of an action is a random variable that depends on the action and the true (hidden) hypothesis. In prior work, the probability distribution functions on the outcomes are assumed to be known. In the present work we assume that these distributions are not known, and only some rough information about the outcomes of the actions is known, to be made more precise further on.

Active hypothesis testing is an increasingly important problem these days, with applications that include the following. (a) Medical diagnostics ([2]) systems that include clinical trials for testing a new treatment, or diagnostics of a new disease. (b) Crowdsourcing: online platforms for task-worker matching such as Amazon's Mechanical Turk or TaskRabbit, where, as new tasks arrive, they must be matched to workers capable of working on them. (c) Customer hotline centres or Q&A forums: online platforms such as StackExchange where questions are submitted, and users with varying capabilities are available for providing an answer. This includes customer service centres where customer tickets are submitted and the nature of the problem must be learned before its treatment (an example where supervised learning techniques are used is [3]). (d) Content search problems where an incoming image must be matched to known contents, as studied in [4].

We now informally describe our model. In the general instance of our problem, the true hypothesis, $\theta^*$ is one in a set of $J$ hypotheses, $\mathcal{J} = \{\theta_1, \ldots, \theta_J\}$, and a set of $W$ actions is available, where the outcomes of the actions depend on the true hypothesis. When the true hypothesis is $\theta_j$ and

action $w$ is chosen, a noisy outcome $X_{w,j} \in \mathcal{J}$ is observed, whose distribution, $p_{w,j}(\cdot) \in \mathbb{P}(\mathcal{J})$, is given. The objective then is to select an action at each step so as to infer the true hypothesis in a minimum number of steps, with a given accuracy. In our model, we assume that the decision maker has limited information about the outcome distributions. We define the *principal* set of an action $w$ as $\mathcal{J}_w \subseteq \mathcal{J}$. When action $w$ is sampled, a noisy binary outcome $y \in \{-1, 1\}$ is observed, which gives an indication on whether the action classifies the hypothesis in the set $\mathcal{J}_w$. The *quality* of action $w$, $\alpha_w$ is related to the noise in the outcome. Rather than the distributions $p_{w,j}(\cdot)$, we assume that the decision maker only has knowledge of the principal set $\mathcal{J}_w$ and quality $\alpha_w$ of each action.

## 1.1 Related work

Since the seminal work by Chernoff [1], active hypothesis testing and variants of the problemhave been studied through various perspectives (see [5] for a brief survey). Chernoff derived a simple heuristic algorithm whose performance is shown to achieve asymptotic optimality in the regime where the probability of error vanishes. Specifically, it is shown that as the probability of error $\delta$ decreases the expected number of samples needed by Chernoff's algorithm grows as $-\log(\delta)$. Most of the past literature in active sequential hypothesis testing has dealt with extensions of Chernoff's model, and has shown that Chernoff's algorithm performs well in more general settings [6, 7]. A notable exception is [8], where the impact of the number of hypotheses is analyzed and an algorithm that performs better than Chernoff's benchmark is provided for the case of large values of $J$.

Our work differs from prior work in a few ways. First, the hypothesis need not be *locally identifiable*. While in [1] each action is able to distinguish each pair of hypotheses, we assume that each hypothesis is *globally identifiable*, i.e., each pair of hypotheses can be discerned by at least one action. This is a common assumption in the area of distributed hypothesis testing ([9, 10]) and a weaker assumption than that of Chernoff. Note that dropping this assumption is not novel in itself, and has been done in other work such as [8]. Second, a novel extension in our work, differing from [8] is that we do not assume full knowledge on the actions' statistical parameters. The responses of actions are noisy, and in past literature the probability distributions governing them was assumed to be known. In our model, we drop this assumption, and we only require to know a lower bound $\alpha_w > 1/2$ on the probability that action $w$ will provide a correct response, no matter the hypothesis we want to test. As far as we know, no previous work in active sequential learning has tackled the problem of incomplete statistical information and we believe that such an extension may provide a non-negligible impact in real-life applications.

Active hypothesis testing is similar to the problem of Bayesian active learning. This latter perspective in considered in [11] where noisy Bayesian active learning setting is used on the hypothesis testing problem with asymmetric noise and a heuristic based on the extrinsic Jensen-Shannon (EJS) divergence [12] is proposed. As in [8], full knowledge of the probability distributions governing the noise is available. In contrast, in our work we consider a more restricted model where, only a binary outcome with noise is given by the actions on the large hypothesis space. Inference with binary responses is considered in work on generalized binary search (GBS) [13], which is special case where the label set (outcome of actions) is binary with the case of symmetric, non-persistent noise. Our work differs from this type of work in that we consider asymmetric label-dependent noise, that is, $\alpha_w$ varies with action $w$.

We thus position our work between [11, 8] and [13]. While the former assumes full knowledge on the noise distributions, we assume that only a binary response is provided and only a lower bound on the value that governs the outcome is known, and while the latter considers symmetric noise, we extend to asymmetric label-dependent noise.

**Our contribution.**   Our main objective is to investigate the minimum sample query size of this system for a certain level of accuracy in the inference of the true hypothesis, and to design efficient policies for this inference. Our contributions in the present paper are as follows. First, we consider the system under limited knowledge of outcome distribution. This restricted scenario adds a significant constraint for the action selection policy, and the belief vector update policy. To the best of our knowledge, this restricted scenario has not been considered in past literature. Second, under the limited knowledge constraint, we propose the Incomplete-Bayesian Adaptive Gradient (IBAG) policy which includes a belief vector update rule that we call Incomplete-Bayesian, and an action selection rule, named Adaptive Gradient, that follows the drift of the (unknown) coordinate of interest in the

belief vector. Third, we derive a lower bound on the sample size for the system under incomplete information, and show that the performance of IBAG matches this bound. We also carry out numerical experiments to compare IBAG to prior work.

## 2  Model

The classic model of the active sequential learning problem consists in sequentially selecting one of several available sensing actions, in order to collect enough information to identify the true hypothesis, as considered in [1]. We thus consider a system where a decision maker has at his disposal a finite set of actions $\mathcal{W} = \{1, \ldots, W\}$, and there are a set of $J = |\mathcal{J}| < \infty$ possible hypothesis, $\mathcal{J} = \{\theta_1, \ldots, \theta_J\}$. (For the rest of the paper, we refer to a hypothesis only by its index, i.e., $j$ for hypothesis $\theta_j$, for ease of notation.) When the true hypothesis is $j$ and action $w$ is sensed, the outcome $X_{w,j} \in \mathcal{J}$ is sampled from the distribution $p_{w,j}(\cdot) \in \mathbb{P}(\mathcal{J})$, i.e., $\mathbb{P}\{X_{w,j} = j'\} = p_{w,j}(j')$.

In our model, we assume to have limited information about the actions and this affects the classic model in two ways. First, for every sampled action $w$, a binary outcome $y \in \{-1, 1\}$ is observed, indicating whether the inference of hypothesis by this action is in $\mathcal{J}_w$ or not, i.e., the response observed is $Y_{w,j} \in \{-1, 1\}$ where

$$Y_{w,j} = \begin{cases} 1, & \text{if } X_{w,j} \in \mathcal{J}_w, \\ -1, & \text{if } X_{w,j} \notin \mathcal{J}_w. \end{cases}$$

The subset $\mathcal{J}_w \subseteq \mathcal{J}$ is assumed to be known, and it is described by the matrix $g \in \{-1, 1\}^{W \times J}$ where

$$g_{w,j} = \begin{cases} 1, & \text{if } j \in \mathcal{J}_w, \\ -1, & \text{if } j \notin \mathcal{J}_w. \end{cases} \tag{1}$$

Observe that the probability an action $w$ correctly identifies the subset to which the true hypothesis $j$ belongs is given by $q_{w,j} := \mathbb{P}\{Y_{w,j} = g_{w,j}\} = \sum_{j':g_{w,j}=g_{w,j'}} p_{w,j}(j')$. However, as a second restriction, instead of knowing $q_{w,j}$, the capacity, or quality, of an action $w$ is captured by $\alpha_w$ where we assume that

$$q_{w,j} \geq \alpha_w, \qquad \forall\, j \in \mathcal{J},\ w \in \mathcal{W}. \tag{2}$$

We thus characterize each action by its *principal set*, $\mathcal{J}_w$, and its *quality*, $\alpha_w$.

**Assumption 1.** *For every action $w \in \mathcal{W}$, the principal sets $\mathcal{J}_w \subseteq \mathcal{J}$ and the quality $\alpha_w \in (1/2, 1)$ are known. Denote by $\Delta_w = 2\alpha_w - 1$ where $\Delta_w \in [\Delta^m, \Delta^M]$ and $\Delta^m, \Delta^M \in (0, 1)$.*

Since each action can only indicate whether the hypothesis belongs to a subset or not, there must exist an action $w \in \mathcal{W}$ for which $j_1$ and $j_2$ belong to different subsets, for all pairs $j_1, j_2 \in \mathcal{J}$. Define the subset $\mathcal{W}_{j_1,j_2} \subseteq \mathcal{W}$ as $\mathcal{W}_{j_1,j_2} = \{w \in \mathcal{W} : g_{w,j_1} g_{w,j_2} = -1\}$.

**Assumption 2.** *For every $j_1, j_2 \in \mathcal{J}$, the subset $\mathcal{W}_{j_1,j_2}$ is nonempty, i.e., each hypothesis is* globally *identifiable.*

For every action $w \in \mathcal{W}$ and hypothesis $j \in \mathcal{J}$ we define the subsets $\mathcal{J}_{w,+j}$ and $\mathcal{J}_{w,-j}$ which are, respectively, given by the hypotheses that action $w$ cannot and can distinguish from $j$, i.e.,

$$\mathcal{J}_{w,+j} = \{j' \in \mathcal{J} : g_{w,j'} g_{w,j} = 1\}, \qquad \mathcal{J}_{w,-j} = \{j' \in \mathcal{J} : g_{w,j'} g_{w,j} = -1\}.$$

Note that $w \in \mathcal{W}_{j_1,j_2}$ if and only if $j_2 \in \mathcal{J}_{w,-j_1}$ (or equivalently $j_1 \in \mathcal{J}_{w,-j_2}$).

We aim to design a simple algorithm to infer the correct hypothesis using as few actions as possible. The true hypothesis will be denoted by $j^* \in \mathcal{J}$. The learning process is captured by the evolution of the *belief vector* $\boldsymbol{\nu}(t) \in \mathbb{P}(\mathcal{J})$, where $\nu_j(t)$ denotes the decision maker's confidence at time $t$ that the true hypothesis is $j$. At the initial step $t = 1$, the belief vector $\boldsymbol{\nu}(1) \in \mathbb{P}(\mathcal{J})$ is initialized so that $\nu_j(1) > 0, j \in \mathcal{J}$. Since we assume to initially lack any information on the true hypothesis, without loss of generality, we set $\nu_j(1) = 1/J$ for every $j \in \mathcal{J}$.

At every step $t \geq 1$, according to the belief vector $\boldsymbol{\nu}(t)$, the decision maker determines the next action to sense $F_W(\boldsymbol{\nu}(t)) = w(t) \in \mathcal{W}$ according to some selection rule $F_W(\cdot)$. The outcome $y(t) \in \{-1, 1\}$ from the chosen action $w(t)$ is used to update the belief vector according to an update rule $\boldsymbol{F}_U(\boldsymbol{\nu}(t), w(t), y_{(t)}) = \boldsymbol{\nu}(t + 1) \in \mathbb{P}(\mathcal{J})$. The algorithm ends at time $T^*$, and the

inferred hypothesis is given by $\hat{j} = \arg\max_{j\in\mathcal{J}} \nu_j(T^*)$. Sensing actions is stopped when one of the posteriors is larger than $1 - \delta$, for some $\delta > 0$:

$$T^* = \inf_{t\geq 0}\{\max_{j\in\mathcal{J}} \nu_j(t) > 1 - \delta\}. \tag{3}$$

## 3 The Incomplete-Bayesian update rule

We now describe how the decision maker updates the belief vector after he observes the outcome of an action. Given a belief vector $\boldsymbol{\nu} \in \mathbb{P}(\mathcal{J})$ and the observation $y \in \{-1, 1\}$ obtained from action $w \in \mathcal{W}$, define

$$\tilde{f}(y, j, w) = \begin{cases} q_{w,j}, & y = g_{w,j}, \\ 1 - q_{w,j}, & y = -g_{w,j}, \end{cases} \qquad f(y, j, w) = \begin{cases} \alpha_w, & y = g_{w,j}, \\ 1 - \alpha_w, & y = -g_{w,j}. \end{cases}$$

Note that $\tilde{f}(y, j, w)$ denotes the probability of having outcome $y$ given that the action $w$ is chosen and the true hypothesis is $j$. The standard Bayesian update rule is given by the map $\boldsymbol{F}_U^B(\boldsymbol{\nu}, w, y)$, where $F_{U,j}^B(\boldsymbol{\nu}, w, y) = \frac{\tilde{f}(y,j,w)\nu_j}{\sum_{i\in\mathcal{J}} \tilde{f}(y,i,w)\nu_i}$. In our model, however, the values $q_{w,j}$ for $w \in \mathcal{W}$ are unknown to the decision maker. Hence, we introduce the *Incomplete Bayesian* (IB) update rule, which mimics the Bayesian rule, but with limited knowledge on outcome probabilities. The IB update rule is given by the map $\boldsymbol{F}_U(\boldsymbol{\nu}, w, y)$, where

$$F_{U,j}(\boldsymbol{\nu}, w, y) = \frac{f(y, j, w)\nu_j}{\sum_{i\in\mathcal{J}} f(y, i, w)\nu_i}. \tag{4}$$

Observe that Bayesian and IB update rules are identical when $q_{w,j} = \alpha_w$.

In practice, the $\nu_j(t)$ evolves according to both the quality of the chosen action, $\alpha_w$, and the relation between this action's principal set $\mathcal{J}_w$ and the current state of the belief vector $\boldsymbol{\nu}(t)$. This dependence is formalized in the following lemma whose proof is included in the supplementary material, Section B.

**Lemma 1.** *Given $\boldsymbol{\nu}(t) \in \mathbb{P}(\mathcal{J})$ and $w(t) \in \mathcal{W}$, then it holds that*

$$\frac{\nu_{j^*}(t+1)}{\nu_j(t+1)} = \frac{\nu_{j^*}(t)}{\nu_j(t)} \times \begin{cases} 1, & w.p. \\ indic1\{w(t) \notin \mathcal{W}_{j^*,j}\}, & \\ \frac{1+\Delta_{w(t)}}{1-\Delta_{w(t)}}, & w.p.\ \mathbb{1}\{w(t) \in \mathcal{W}_{j^*,j}\}q_{w(t),j^*}, \\ \frac{1-\Delta_{w(t)}}{1+\Delta_{w(t)}}, & w.p.\ \mathbb{1}\{w(t) \in \mathcal{W}_{j^*,j}\}(1 - q_{w(t),j^*}). \end{cases}$$

### 3.1 A lower bound on the sample size

Note that the IB update rule alone sets some constraints on the performance. In particular, if we require the error probability to be low, then the expected number of samples is necessarily larger than a certain quantity depending on the model parameters. We show that this quantity asymptotically grows as $-\log\delta$ in the asymptotic regime where $\delta \to 0$.

**Theorem 1.** *Assume the IB update rule is applied to the belief vector and that*

$$\lim_{\delta\to 0} \mathbb{P}\{\nu_{j^*}(T^*) \leq \delta\} \leq \tilde{\gamma} < 1.$$

*Then, there exist functions $K_0^l(\delta)$, $K_1^l(\delta)$ such that*

$$\mathbb{E}[T^*] \geq K_1^l(\delta)\log\frac{1}{\delta} + K_0^l(\delta), \qquad \lim_{\delta\to 0} K_i^l(\delta) \geq K_i^l > 0, \quad for\ i = 0, 1.$$

The proof of this result is presented in the supplement, Section A.2. We sketch the proof here. We first define

$$S_t(j_1, j_2) = \log\frac{\nu_{j_1}(t)}{\nu_{j_2}(t)}, \qquad S(j_1, j_2) = S_{T^*}(j_1, j_2),$$

and show that, on the one hand, if $\mathbb{P}\{\hat{j} \neq j^*\}$ is small, then $\sum_{j\neq j^*} S(j^*, j)$ is large with high probability, and on the other hand, if $t$ is small, then $\sum_{j\neq j^*} S_t(j^*, j)$ is small with high probability.

We use these properties to derive a lower bound on the tail probability of $T^*$, and thus on its expected value.

Further, we can control the belief vector evolution by deriving bounds on the ratio between coordinates of the belief vector under the IB policy. Specifically, in the supplementary material Section A.3, we bound the probability that $\nu_j(t) > \nu_{j^*}(t)$ at a certain time, and investigate how this probability evolves with $t$.

## 4 Adaptive Gradient: the action selection policy

### 4.1 A gradient-based selection policy

We now present an action selection policy that, together with the IB update rule, defines our active learning algorithm, which we call the Incomplete-Bayesian Adaptive Gradient (IBAG) policy. We will then analyze the complete algorithm showing that its performance asymptotically matches the lower bound provided in Theorem 1 as $\delta \to 0$.

We focus on the $j^*$-th coordinate of the belief vector, and define the drift at time $t$ as

$$D_w(\boldsymbol{\nu}(t)) = \mathbb{E}[\nu_{j^*}(t+1)|\boldsymbol{\nu}(t), w(t) = w] - \nu_{j^*}(t).$$

Simple algebra and (4) yield the following Lemma.

**Lemma 2.** *It holds that*

$$D_w(\boldsymbol{\nu}(t)) = 4\Delta_w \nu_{j^*}(t)\nu_{w,-j^*}(t)\left(\frac{q_{w,j^*} - \alpha_w + \Delta_w \nu_{w,-j^*}(t)}{1 - \Delta_w^2\left(1 - 2\nu_{w,-j^*}(t)\right)^2}\right), \tag{5}$$

*where*

$$\nu_{w,+j} = \sum_{j \in \mathcal{J}_{w,+j}} \nu_j, \qquad \nu_{w,-j} = \sum_{j \in \mathcal{J}_{w,-j}} \nu_j.$$

Assume for a moment that we know the true hypothesis $j^*$ and $q_{w,j^*}$ for every $w \in \mathcal{W}$. Then, in order to let $\nu_{j^*}(t)$ grow as much as possible, we would greedily select the action $w$ which maximizes $D_w(\boldsymbol{\nu}(t))$. Our worker selection policy will attempt to mimic as closely as possible this greedy policy, while operating without complete information.

**Lemma 3.** *It holds that $D_w(\boldsymbol{\nu}(t)) \geq D_w^L(\boldsymbol{\nu}(t))$, where*

$$D_w^L(\boldsymbol{\nu}(t)) = 4\nu_{j^*}(t)\frac{\Delta_w^2 \nu_{-w}^2(t)}{1 - \Delta_w^2\left(1 - 2\nu_{-w}(t)\right)^2}, \tag{6}$$

*and*

$$\nu_{-w}(t) = \min\left\{\sum_{j \in \mathcal{J}_w} \nu_j(t), \sum_{j \notin \mathcal{J}_w} \nu_j(t)\right\}$$

The proof follows from the fact that $D_w(\boldsymbol{\nu}(t))$ is increasing both in $q_{w,j^*}$ and $\nu_{w,-j^*}(t)$ for every $w \in \mathcal{W}$, and the observation that that $q_{w,j^*} \geq \alpha_w$ and $\nu_{w,-j^*}(t) \geq \nu_{-w}(t)$.

Note that $D_w^L(\boldsymbol{\nu}(t))$ provides us a tight lower bound on the expected growth of the coordinate of the true hypothesis if action $w$ is chosen at step $t$. Indeed, $D_w^L(\boldsymbol{\nu}(t))$ can be decomposed to a part that uses the $j^*$-th coordinate of the belief vector and a part than can be computed without knowing $j^*$. The Adaptive Gradient (AG) selection rule, then chooses at step $t$, the action $w^D(t) \in \mathcal{W}$ such that

$$w^D(t) = F_W(\boldsymbol{\nu}(t)) = \arg \max_{w \in \mathcal{W}} G(\nu_{-w}, \Delta_w), \qquad G(v, d) = \frac{d^2 v^2}{1 - d^2\left(1 - 2v\right)^2}, \tag{7}$$

i.e., we select the action maximizing the current lower bound on the expected growth of the $j^*$-coordinate of the belief vector. Ties are broken uniformly.

*Remark:* Assume the actions have different costs of sensing. The AG selection rule can then be generalized as follows:

$$w^D(t) = F_W^c(\boldsymbol{\nu}(t)) = \arg \max_{w \in \mathcal{W}} \frac{G(\nu_{-w}, \Delta_w)}{c_w}. \tag{8}$$

## 4.2 An upper bound

We now present our main result. We show that the expected number of samples required by our algorithm IBAG asymptotically matches the lower bound obtained in Theorem 1.

**Theorem 2.** *Under the* IBAG *algorithm, there exist constants* $K_0^u$, $K_1^u > 0$ *independent of* $\delta$ *such that*

$$\mathbb{E}[T^*] \leq K_1^u \log \frac{1}{\delta} + K_0^u.$$

The proof is provided in supplementary material, Section A.5. This result is based on the intuition that IBAG never selects an action that is too uninformative relative to the other actions. Specifically, the information provided by an action $w$ at time $t$ depends on its quality $\alpha_w$ and outcome over the subset $\mathcal{J}_{w,-j^*}$. In other words, the value $\nu_{w,-j^*}$ must decrease to 0, hence the higher this value is for a given action $w$, the more we can still learn from sensing this action. As a proxy for $\nu_{w,-j^*}$ we use $\nu_{-w}$ which also must be as large as possible. The following lemma, whose proof is given in supplementary material, Section B, provides bounds on the relative quality of $\nu_{-w^D(t)}$ compared to $\nu_{-w}$.

**Lemma 4.** *For every* $w \in \mathcal{W}$, *it holds that* $\nu_{-w^D(t)} \geq \frac{\Delta^m}{\Delta^M} \nu_{-w}$.

# 5 Numerical results

We now present numerical results based on simulations. In order to gain practical insight, we will focus on a task labelling application. A task labelling problem might arise in a crowdsourcing scenrio such as Amazon's Mechanical Turk or Content search problems where an incoming image must be matched to known contents. The mapping to the hypothesis testing problem is as follows. The set of hypotheses $\mathcal{J}$ corresponds to the set of task labels, with $j^*$ the true hypothesis being the latent task label that must be inferred. The set of $W$ actions corresponds to $W$ workers who perform the labelling when sampled, where $p_{w,j}(j')$ is the probability that worker $w$ assigns the task the label $j'$ when the true label is $j$. For each worker $w$, we will call $\mathcal{J}_w$ the expertise of the worker (principal set of the actions), and $\alpha_w$ will be the quality of the worker. We will first investigate the impact of the lack of exact knowledge, i.e., the difference between $\alpha_w$ and $q_{w,j}$, that we call *slack*. We then compare our algorithm to that in [1] and that of [13] for a few scenarios of interest.

## 5.1 The effect of the slack

Here we present a simulated scenario with $J = 100$, $W = 15$, and fixed subsets $\{\mathcal{J}_w\}_{w \in \mathcal{W}}$ satisfying Assumption 2. We set $\delta \approx 0.001$, and assume the incoming job-type to be $j^* = 1$. In Figure 1 we present the results of 1000 runs of the simulation for every instance of respectively the first and second scenario described below. Recalling that the simulation stops as soon as $\max_j \nu_j(t) > 1 - \delta$, we specify that out of the entire set of simulations of these scenarios the algorithm never failed to infer the correct incoming job type $j^* = 1$. For both scenarios, in Figure 1(left) we display the averaged sample paths of the coordinate $\nu_{j^*}(t)$ and in Figure 1(right) the average sample size required for the decision maker to make an inference.

**The performance upper bound is pessimistic.** In the first set of simulations, scenario A, we fix the quality vector $\alpha$ with $\alpha_w \in (0.55, 0.6)$ for every worker $w \in \mathcal{W}$. We then let the parameter $s$ vary in $\{0, .05, .1, .15, .2, .25, .3\}$ and assume $q_{w,j^*} = \alpha_w + s$ for every $w \in \mathcal{W}$. In Theorem 2 we proved an upper bound for $\mathbb{E}[T^*]$ when the IBAG algorithm is employed. It can be observed that the upper bound does not depend on $q_{w,j^*}$, but only on $\alpha_w$. In fact, the upper bound is obtained by looking at the worst case scenario, where $q_{w,j^*} = \alpha_w$ for every $w \in \mathcal{W}$ and $j \in \mathcal{J}$. As the slack $s$ grows, the performance of the algorithm drastically improves even if it is not reflected in the upper bound term.

**Robustness to perturbations in estimate of worker skills.** In the second set of simulations, scenario B we fix the quality vector $q_{w,j^*} \in (0.85, 0.9)$ for every worker $w \in \mathcal{W}$. We then let the parameter $s$ vary in $\{0, .05, .1, .15, .2, .25, .3\}$ and set $\alpha_w = q_{w,j^*} - s$ for every $w \in \mathcal{W}$. It is observed that the IBAG algorithm performs well even when the decision maker's knowledge of the skills is not precise, and he decides to play safe by reducing the lower bound $\alpha(w)$.

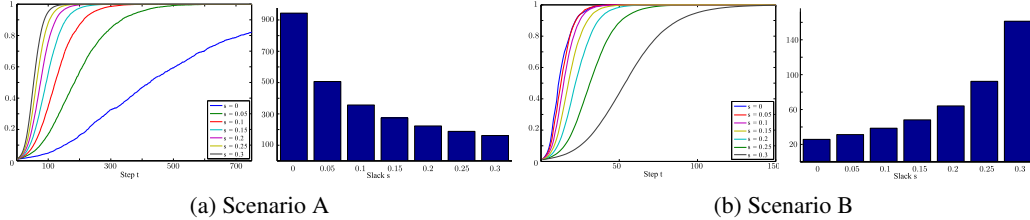

(a) Scenario A          (b) Scenario B

Figure 1: ((a), (b) left) Empirical average of the sample paths of the process $\nu_{j^*}(t)$, ((a), (b) right) Empirical average of the sample size $T^*$.

We therefore deduce that the learning process *strongly* depends on the true skills of the worker $q_{w,j}$ (Figure 1(a)), however their exact knowledge is *not fundamental* for IBAG to behave well (Figure 1(b)) - it is robust to small perturbations.

## 5.2 Comparison to existing algorithms

**Chernoff algorithm.** As we mentioned, most of the existing sequential hypothesis testing algorithms are based on Chernoff's algorithm presented in [1]. Such an algorithm, at step $t$ identifies the job-types $j_1, j_2 \in \mathcal{J}$ associated with the two highest values of $\boldsymbol{\nu}(t)$ and selects the class of workers $w^C$ that best distinguishes $j_1$ and $j_2$, i.e., $w^C = \arg\max_{w \in \mathcal{W}_{j_1,j_2}} \alpha_w$. In the asymptotic regime with $\delta \to 0$, the expected sample size required by the Chernoff's algorithm is of order $-\log \delta$, exactly as with IBAG. This has been proven ([1, 8]) in the case with full knowledge of the matrix $p_{w,j}(\cdot)$. What we emphasize here is that by focusing only on the two highest components of $\boldsymbol{\nu}(t)$, the decision maker loses information that might help him make a better selection of worker $w(t)$. In particular, Chernoff's algorithm bases its decision largely on the workers' skills and thus does not behave as well as it should when these are not informative enough.

**Soft-Decision GBS algorithm.** The algorithm proposed in [13] generalizes the intuition behind optimal GBS algorithms in noiseless environments. This algorithm, given a belief vector $\boldsymbol{\nu}(t)$ at step $t$ picks the worker $\bar{w}$ such that $\bar{w} = \arg\min_w \left| \sum_{j \in \mathcal{J}} \nu_j g_{w,j} \right| = \arg\min_w \left| \sum_{j \in \mathcal{J}_w} \nu_j - \sum_{j \notin \mathcal{J}_w} \nu_j \right| = \arg\max_w \{\nu_{-w}\}$. Intuitively, the Soft-Decision GBS algorithm selects the worker that is the most "unsure", in the sense that the worker splits the belief vector as evenly as possible. Since the model in [13] does not allow for different qualities of the workers (noise is symmetric there), this feature does not play a role on the worker selection policy. Note that when the quality of all workers are identical, the Soft-Decision GBS and the IBAG algorithms are identical. In [13], an asymptotic performance analysis is presented, and under certain constraints on the problem geometry, it is shown that the sample size required is of order $-\log \delta + \log J$, and once again the performance in terms of the error probability matches with IBAG.

We now compare our algorithm IBAG with the Chernoff algorithm under three scenarios and with Soft-Decision GBS only for the third scenario where the quality $\alpha_w$ or workers (noise in GBS) differ among the workers.

In the first scenario, we set $J = 32$, $j^* = 1$, and $\delta = 0.003$. We assume two kinds of worker classes. We have 5 'generalist' workers, each of whom has $|\mathcal{J}_w| = J/2 = 16$ and moreover for every pair of job types $(j_1, j_2)$ there exists a generalist belonging to $\mathcal{W}_{j_1,j_2}$. In addition, we have 32 'specialist' workers who can distinguish exactly 1 job-type, i.e., $|\mathcal{J}_w| = 1$. We assume that there is one specialist per job-type, and note that among them there is also $w^*$ such that $\mathcal{J}_{w^*} = \{j^*\}$. We consider two cases: in case A, the skills of the workers are identical, $\alpha_w = 0.8$ for every $w \in \mathcal{W}$, and in case B we drop the generalists' skill level to $\alpha_w = 0.75$. We assume $q_{w,j} = \alpha_w$ for every $w \in \mathcal{W}$ and $j \in \mathcal{J}$.

In the second scenario, we set $J = 30$ with only specialists present. We set $\delta = 0.003$ and $j^* = 1$. In this scenario we consider two cases as well, in case A $\alpha_w = 0.7$ for every worker, while in case B we drop the skill level of the specialist on job-type $j^*$ to 0.65, representing a situation where the system is ill-prepared for an incoming job. We assume $q_{w,j} = \alpha_w$ for every $w \in \mathcal{W}$ and $j \in \mathcal{J}$.

We display the results for both scenarios in Figure 2. In Figure 2(top) we display boxplots of the number of queries required and in Figure 2(bottom) we show the expectation of the number of queries per kind of worker. In both scenarios, the performance of Chernoff's algorithm is drastically

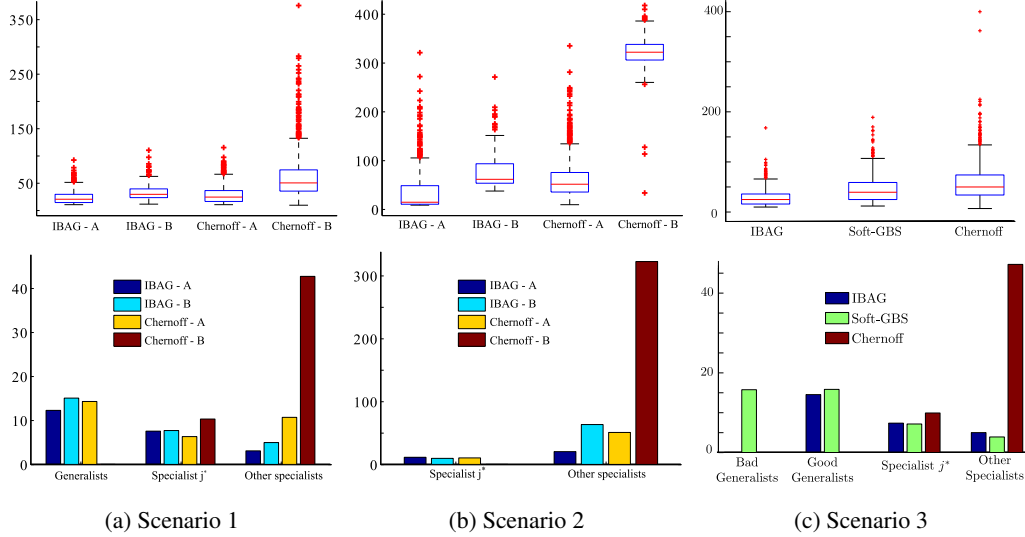

Figure 2: (top) Boxplot of the sample size $T^*$. (bottom) Empirical expected number of times the different groups of workers are queried.

weakened by only a tiny variation in $\alpha_w$, yielding a very different behavior. In the first scenario, although it is very informative to query the generalists in an early explorative stage, under Chernoff's algorithm the selection of the workers relies too much on the skill levels and therefore always queries the specialists. The IBAG algorithm, on the other hand, sensibly decides at each step on the trade-off between getting rough information on a larger set of job pairs, or getting more precise information on a smaller set, and seems to better grasp this *quality vs quantity* dilemma.

Similarly, in case B of the second scenario, the low-quality workers (the specialist in $j^*$) are never selected by Chernoff's algorithm, even if their responses have a large impact on the growth of $\nu_{j^*}(t)$. For both cases A and B we see that IBAG outperforms Chernoff.

In the third scenario we set $J = 32$, $W = 42$, and $\delta = 0.03$. We have five low-quality generalist workers with $\alpha_w = 0.55$, five high-quality generalist workers with $\alpha_w = 0.75$. The remaining 32 workers are specialists with $\alpha_w = 0.8$. The plots comparing all three algorithms is shown in Figure 2(iii). We observe again that the Chernoff algorithm never queries generalists and performs the worst. IBAG outperforms Soft-GBS because it queries high-quality workers preferentially while Soft-GBS doesn't consider quality.

## 6 Discussion and conclusion

We have presented and analyzed the IBAG algorithm, an intuitive active sequential learning algorithm which requires only a rough knowledge of the quality and principal set of each available action. The algorithm is shown to be competitive and in many cases outperforms Chernoff's algorithm, the benchmark in the area.

As far as we know, this is the first attempt to analyze a scenario where the decision maker has limited knowledge of the system parameters. In Section 5 we studied through simulations, the effect of this lack of exact knowledge on the performances of the system, in order to quantify the tradeoff between *caution*, i.e., how close $\alpha_w$ is to $q_{w,j}$, and the *cost*. The numerical analysis suggests that a moderate caution does not worsen drastically the performance. In the supplement Section C we analyze formally this tradeoff and show results on how cautious the decision maker can be while still ensuring good performance.

A further element of incomplete knowledge would be to allow slight perturbations on the principal sets of the actions. In the present paper we have assumed to know with certainty, for every $w \in \mathcal{W}$ and $j \in \mathcal{J}$, whether $w$ has $j$ in its principal set ($j \in \mathcal{J}_w$), or not. In future work we will investigate the impact of uncertainty in the expertise, for instance having $j \in \mathcal{J}_w$ with some probability $p_{j,w}$.

As a last remark, it would be interesting to analyze the model when the different actions have heterogeneous costs. Note that the IBAG algorithm naturally extends to such case, as mentioned in equation (8). The IBAG algorithm in the framework of the task-worker system could give definitive answers on whether it is better to sample a response from a cheap worker with a general expertise and low skill or from more expensive workers with narrow expertise and higher skill.

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
