[Supplementary Material · nips_2017_supp.pdf]

# Supplementary material for "Adaptive Active Hypothesis Testing under Limited Information"

## Abstract

This supplementary document contains the proofs of propositions, lemmas, and theorems.

## A   Extended proofs

### A.1   Additional notation

In the following proofs we will use some additional notation which we introduce here. Define the random variables

$$Z_s(j_1, j_2) = \log H_s(j_1, j_2), \qquad H_s(j_1, j_2) = \frac{f(Y(s), j_1, w(s))}{f(Y(s), j_2, w(s))}$$

so as that

$$S_t(j_1, j_2) = \log \frac{\nu_{j_1}(t)}{\nu_{j_2}(t)} = \sum_{s<t} Z_s(j_1, j_2),$$

Let $V(j_1, j_2)$ denote the variance of $Z_s(j_1, j_2)$, and

$$V = \max_{j_1, j_2 \in \mathcal{J}} V(j_1, j_2).$$

Note that $\Delta_w \leq \Delta^M < 1$, implies that $V < \infty$.

The quantity $I(j^*)$ captures the *maximal information* attainable from the system when the incoming job type is $j^*$,

$$I(j^*) = \max_{w \in \mathcal{W}} \{ \sum_{j \in \mathcal{J}} I(j^*, j, w) \}, \qquad I(j^*, j, w) = \begin{cases} \Delta_w \log \left( \frac{1+\Delta_w}{1-\Delta_w} \right), & j \in \mathcal{J}_{w,-j^*}, \\ 0, & j \in \mathcal{J}_{w,+j^*}. \end{cases}$$

The term $\sigma(j^*)$ is the *maximal slack* given that the incoming job type is $j^*$, and denotes how accurately $\alpha_w$ approximates $q_{w,j^*}$, specifically

$$\sigma(j^*) = \max_{w \in \mathcal{W}} \{ q_{w,j^*} - \alpha_w \}.$$

### A.2   Proof of Theorem 1

As stated in Section 3, the proof is based on showing that

- If $\mathbb{P}\{\hat{j} \neq j^*\}$ is small, then $\sum_{j \neq j^*} S(j^*, j)$ is large with high probability.

17      • If $t$ is small, then $\sum_{j \neq j^*} S_t(j^*, j)$ is small with high probability.

18  These properties will be shown respectively in Proposition 1(S) and 2(S).

19  **Proposition 1** (S). *It holds that*

$$\mathbb{P}\Big\{ \sum_{j \neq j^*} S(j^*, j) < (J-1) \log \frac{1-\delta}{\delta} \Big\} \leq \mathbb{P}\{\nu_{j^*}(T^*) \leq \delta\} =: \gamma(\delta).$$

20  *Proof.* Denote the event $B = \Big\{ \sum_{j \neq j^*} S(j^*, j) < (J-1) \log \frac{1-\delta}{\delta} \Big\}$, and we aim to show that

21  $\mathbb{P}\{B | \hat{j} = j^*\} = 0$. It holds that

$$\hat{j} = j^* \quad \Rightarrow \quad \frac{\nu_{j^*}(T^*)}{\nu_j(T^*)} > \frac{1-\delta}{\delta}, \quad \forall j \neq j^*$$

$$\Rightarrow \quad S_{j^*, j} > \log \frac{1-\delta}{\delta}, \quad \forall j \neq j^*$$

$$\Rightarrow \quad \sum_{j \neq j^*} S_{j^*, j} > (J-1) \log \frac{1-\delta}{\delta} \Rightarrow \bar{B},$$

22  and therefore $\mathbb{P}\{B | \hat{j} = j^*\} = 0$. We conclude, since

$$\mathbb{P}\{B\} = \mathbb{P}\{B | \hat{j} \neq j^*\} \mathbb{P}\{\hat{j} \neq j^*\} + \underbrace{\mathbb{P}\{B | \hat{j} = j^*\}}_{=0} \mathbb{P}\{\hat{j} = j^*\}$$

$$= \mathbb{P}\{B | \hat{j} \neq j^*\} \mathbb{P}\{\hat{j} \neq j^*\} \leq \mathbb{P}\{\hat{j} \neq j^*\} \leq \mathbb{P}\{\nu_{j^*}(T^*) \leq \delta\}.$$

23                                                                                                          □

24  Before the proof of Proposition 2(S), observe that the notation in Appendix A.1 are required.

25  **Proposition 2** (S). *Given $\epsilon > 0$, it holds that*

$$\mathbb{P}\Big\{ \max_{t \leq T} \sum_{j \neq j^*} S_t(j^*, j) \geq TK(\epsilon) \Big\} \leq \frac{(J-1)^2 V}{T \epsilon^2},$$

26  *for every $T > 0$, where*

$$K(\epsilon) = I(j^*) + 2\sigma(j^*)(J-1) \log \Big( \frac{1 + \Delta^M}{1 - \Delta^M} \Big) + \epsilon.$$

27  *Proof.* Let us rewrite $S_t(j^*, j)$ as follows

$$S_t(j^*, j) = \sum_{s \leq t} \big( Z_s(j^*, j) - \mathbb{E}[Z_s(j^*, j)] \big)$$

$$+ \sum_{s \leq t} \big( \mathbb{E}[Z_s(j^*, j)] - I(j^*, j, w(s)) \big) + \sum_{s \leq t} I(j^*, j, w(s))$$

$$= M_{1,t}(j^*, j) + M_{2,t}(j^*, j) + M_{3,t}(j^*, j),$$

28  and we analyze these three terms separately.

29  The last term, by definition of $I(j^*)$ is such that

$$\sum_{j \neq j^*} M_{3,t}(j^*, j) = \sum_{s \leq t} \sum_{j \neq j^*} I(j^*, j, w(s)) \leq tI(j^*) \leq TI(j^*).$$

30  The second term is such that

$$M_{2,t}(j^*, j) = \sum_{s \leq t} \big( \mathbb{E}[Z_s(j^*, j)] - I(j^*, j, w(s)) \big)$$

$$\leq \sum_{s \leq t} (2q_{w(s), j^*} - 1 - \Delta_{w(s)}) \log \frac{1 + \Delta_{w(s)}}{1 - \Delta_{w(s)}}$$

$$\leq 2t\sigma(j^*) \log \frac{1 + \Delta^M}{1 - \Delta^M} \leq 2T\sigma(j^*) \log \frac{1 + \Delta^M}{1 - \Delta^M}$$

31 hence

$$\sum_{j \neq j^*} M_{2,t}(j^*, j) \leq T(J-1)2\sigma(j^*) \log \frac{1+\Delta^M}{1-\Delta^M}$$

32 It then holds that

$$\sum_{j \neq j^*} M_{1,t}(j^*, j) + M_{2,t}(j^*, j) + M_{3,t}(j^*, j) \geq T K(\epsilon)$$

33 implies that $\sum_{j \neq j^*} M_{1,t}(j^*, j) \geq T\epsilon$, and therefore

$$\mathbb{P}\{\max_{t \leq T} \sum_{j \neq j^*} S_t(j^*, j) \geq T K(\epsilon)\} \leq \mathbb{P}\Big\{ \max_{t \leq T} \sum_{j \neq j^*} M_{1,t}(j^*, j) \geq T\epsilon \Big\}$$

34 Now, observe that $\sum_{j \neq j^*} M_{1,t}(j^*, j)$ is a $\mathcal{L}^2$-martingale. Hence, we can apply Doob's inequality to
35 obtain

$$\mathbb{P}\Big\{ \max_{t \leq T} \sum_{j \neq j^*} M_{1,t}(j^*, j) \geq T\epsilon \Big\} \leq \frac{H}{T^2 \epsilon^2}$$

36 where

$$H = \mathbb{E}\Big[\Big(\sum_{t \leq T} \sum_{j \neq j^*} \big(z_s(j^*, j) - \mathbb{E}[z_s(j^*, j)]\big)\Big)^2\Big] \leq T(J-1)^2 V.$$

37 Hence, we conclude that

$$\mathbb{P}\{\max_{t \leq T} \sum_{j \neq j^*} S_t(j^*, j) \geq T K(\epsilon)\} \leq \frac{(J-1)^2 V}{T \epsilon^2}.$$

38 $\qquad\qquad\qquad\qquad\qquad\qquad\qquad\qquad\qquad\qquad\qquad\qquad\qquad\qquad\qquad\qquad\qquad$ □

39 *Proof.* (of Theorem 1.) Fix $\epsilon > 0$ and define

$$t_\delta = \frac{(J-1)}{K(\epsilon)} \log \frac{1-\delta}{\delta}$$

40 where $K(\epsilon)$ is defined in Proposition 2(S). Due to the law of total probability, it holds that

$$\mathbb{P}\{T^* \leq t_\delta\} \leq \mathbb{P}\{T^* \leq t_\delta, \sum_{j \neq j^*} S(j^*, j) \geq (J-1) \log \frac{1-\delta}{\delta}\}$$
$$+ \mathbb{P}\{\sum_{j \neq j^*} S(j^*, j) < (J-1) \log \frac{1-\delta}{\delta}\}.$$

41 By means of Proposition 2(S), the first term can be bounded as follows

$$\mathbb{P}\{T^* \leq t_\delta, \sum_{j \neq j^*} S(j^*, j) \geq (J-1) \log \frac{1-\delta}{\delta}\} \leq \mathbb{P}\{\max_{t \leq t_\delta} \sum_{j \neq j^*} S_t(j^*, j) \geq t_\delta K(\epsilon)\} \leq \frac{(J-1)^2 V}{t_\delta \epsilon^2},$$

42 where the first inequality follows from the definition of $t_\delta$. Further, the second term can be bounded
43 via Proposition 1(S)

$$\mathbb{P}\{\sum_{j \neq j^*} S(j^*, j) < \log(J-1) \frac{1-\delta}{\delta}\} \leq \mathbb{P}\{\nu_{j^*}(T^*) \leq \delta\} = \gamma(\delta).$$

44 These bounds together yield

$$\mathbb{P}\{T^* \leq t_\delta\} \leq \kappa(\delta), \qquad \kappa(\delta) := \gamma(\delta) + \frac{(J-1)^2 V}{t_\delta \epsilon^2},$$

45  and we conclude by observing that

$$\mathbb{E}[T^*] \geq \mathbb{E}[T^*|T^* > t_\delta]\mathbb{P}\{T^* > t_\delta\}$$
$$\geq t_\delta(1 - \mathbb{P}\{T^* \leq t_\delta\}) \geq t_\delta\big(1 - \kappa(\delta)\big).$$

46  Note that since $\lim_{\delta \to 0} \gamma(\delta) = 0$ and $\lim_{\delta \to 0} t_\delta = \infty$, it holds that

$$\lim_{\delta \to 0} \kappa(\delta) = 0.$$

47  Therefore there exists $\bar{\delta} > 0$ such that for every $\delta < \bar{\delta}$, it holds that $\kappa(\delta) < 1/2$. Hence, for every
48  $\delta < \bar{\delta}$, it holds that

$$\mathbb{E}[T^*] \geq \frac{1}{2}t_\delta = \frac{(J-1)}{2K(\epsilon)}\log\frac{1-\delta}{\delta} \geq \frac{1}{2\big(\Delta_M + 2\sigma(j^*)\big)\log\big(\frac{1+\Delta^M}{1-\Delta^M}\big) + \frac{2\epsilon}{J-1}}\log\frac{1-\delta}{\delta}$$

49  since

$$K(\epsilon) \leq (J-1)\big(\Delta_M + 2\sigma(j^*)\big)\log\big(\frac{1+\Delta^M}{1-\Delta^M}\big) + \epsilon.$$

50  $\square$

## A.3  Control of the belief vector evolution

52  We now control the ratio between coordinates of the belief vector under the IB policy. Specifically, at
53  a certain time $t$, we bound the probability that $\nu_j(t) > \nu_{j^*}(t)$, and investigate how this probability
54  evolves with $t$.

55  The first proposition presents the bound and is based on a coupling argument.

56  **Proposition 3** (S). *Under the* IB *update policy, for every $j \neq j^*$, it holds that*

$$\mathbb{P}\{\frac{\nu_{j^*}(t)}{\nu_j(t)} \leq \epsilon\} \leq \epsilon, \qquad \forall \epsilon > 0, \, t > 0.$$

57  *A Bayesian coupled system.*  We first introduce an alternative way to describe the IB update rule. At
58  time $t$, sample a value $U(t)$ from a uniform random variable in $[0, 1]$. Assume to have chosen the
59  worker-class $w(t) \in \mathcal{W}$, then

$$\nu_j(t+1) = \frac{f(y(t), j, w(t))\nu_j(t)}{\sum_{i \in \mathcal{J}} f(y(t), i, w(t))\nu_i(t)},$$

60  where

$$f(y(t), j, w(t)) = \begin{cases} \alpha_{w(t)}, & y(t) = g_{w(t),j}, \\ 1 - \alpha_{w(t)}, & y(t) = -g_{w(t),j}, \end{cases} \qquad y(t) = \begin{cases} g_{w(t),j^*}, & U(t) < q_{w(t),j^*}, \\ -g_{w(t),j^*}, & U(t) \geq q_{w(t),j^*}. \end{cases}$$

61  We now introduce a coupled belief-process $\boldsymbol{\mu}(t)$, which evolves in parallel with $\boldsymbol{\nu}(t)$ according to the
62  following rule

$$\mu_j(t+i) = \frac{f(y^p(t), j, w(t))\mu_j}{\sum_{i \in \mathcal{J}} f(y^p(t), i, w(t))\mu_i}, \tag{1}$$

63  where

$$y^p(t) = \begin{cases} y(t), & U(t) < \alpha_{w(t)}, \\ -y(t), & U(t) \in [\alpha_{w(t)}, q_{w(t),j^*}), \\ y(t), & U(t) \geq q_{w(t),j^*}. \end{cases} = \begin{cases} g_{w(t),j^*}, & U(t) < \alpha_{w(t)}, \\ -g_{w(t),j^*}, & U(t) \geq \alpha_{w(t)}. \end{cases}$$

64  The peculiarity of the $\boldsymbol{\mu}(t)$ belief vector is that $f(y^p(t), j, w(t))$ is the probability of having response
65  $y^p(t)$ in the *pessimistic* system given where $q_{w,j} = \alpha_w$ for every $w \in \mathcal{W}$ and $j \in \mathcal{J}$. Hence, $\boldsymbol{\mu}(t)$ is
66  updated according to the Bayesian update rule and therefore it represents the *real* posterior probability
67  vector in the pessimistic *fictitious* scenario.

68  This parallel process is introduced due to the following Lemma.

69 **Lemma 1** (S). *If* $\boldsymbol{\nu}(0) = \boldsymbol{\mu}(0)$, *then*

$$\frac{\nu_{j^*}(t)}{\nu_j(t)} \geq \frac{\mu_{j^*}(t)}{\mu_j(t)}, \quad \forall\, t \geq 0.$$

70 *Proof.* Observe that if $w(t) \in \mathcal{W}_{j^*,j}$

$$\frac{\nu_{j^*}(t+1)}{\nu_j(t+1)} = \frac{\nu_{j^*}(t)}{\nu_j(t)} \times \begin{cases} \frac{1+\Delta_{w(t)}}{1-\Delta_{w(t)}}, & \text{if } U(t) \leq q_{w(t),j^*} \\ \frac{1-\Delta_{w(t)}}{1+\Delta_{w(t)}}, & \text{if } U(t) > q_{w(t),j^*} \end{cases}$$

71 and

$$\frac{\mu_{j^*}(t+1)}{\mu_j(t+1)} = \frac{\mu_{j^*}(t)}{\mu_j(t)} \times \begin{cases} \frac{1+\Delta_{w(t)}}{1-\Delta_{w(t)}}, & \text{if } U(t) \leq \alpha_{w(t)} \\ \frac{1-\Delta_{w(t)}}{1+\Delta_{w(t)}}, & \text{if } U(t) > \alpha_{w(t)}. \end{cases}$$

72 on the other hand, if $w(t) \notin \mathcal{W}_{j^*,j}$, it holds that

$$\frac{\nu_{j^*}(t+1)}{\nu_j(t+1)} = \frac{\nu_{j^*}(t)}{\nu_j(t)}, \qquad \frac{\mu_{j^*}(t+1)}{\mu_j(t+1)} = \frac{\mu_{j^*}(t)}{\mu_j(t)}.$$

73 These relations conclude the proof together with the initial condition $\boldsymbol{\nu}(0) = \boldsymbol{\mu}(0)$. $\qquad\square$

74 *Proof of Proposition 3(S).* From Lemma 1(S) it follows immediately that

$$\mathbb{P}\{\frac{\nu_{j^*}(t)}{\nu_j(t)} \leq \beta\} \leq \mathbb{P}\{\frac{\mu_{j^*}(t)}{\mu_j(t)} \leq \beta\},$$

75 and define $B_{t,j^*,j}$ the event

$$B_{t,j^*,j} = \{\frac{\mu_{j^*}(t)}{\mu_j(t)} \leq \beta\}.$$

76 Observe that, over $B_{t,j^*,j}$, it holds that

$$\prod_{s<t} f(y^p(s), j^*, w(s)) \leq \beta \prod_{s<t} f(y^p(s), j, w(s)).$$

77 Hence,

$$\begin{aligned} \mathbb{P}\{B_{t,j^*,j}|j^*\} &= \int_{B_{t,j}} \prod_{s<t} f(y^p(s), j^*, w(s)) \mathrm{d}\Big((y^p(1), w(1)), \ldots, (y^p(t-1), w(t-1))\Big) \\ &\leq \beta \int_{B_{t,j}} \prod_{s<t} f(y^p(s), j, w(s)) \mathrm{d}\Big((y^p(1), w(1)), \ldots, (y^p(t-1), w(t-1))\Big) \\ &= \beta \mathbb{P}\{B_{t,j^*,j}|j\} \leq \beta. \qquad\square \end{aligned}$$

78 This result indicates how likely it is that we are on the wrong path, i.e., $\nu_{j^*}(t)$ should not be lower
79 than $\nu_j(t)$. The next proposition gives us a more explicit bound, however it depends on the sequence
80 of actions chosen up to time $t$.

81 **Proposition 4** (S). *Under the* IB *update policy, for every* $j \neq j^*$, *it holds that*

$$\mathbb{P}\{\frac{\nu_{j^*}(t)}{\nu_j(t)} < M\} \leq (1+M)\big(1 - \Delta_m^2\big)^{|\mathcal{W}_{j^*,j}(t)|/2}, \qquad \forall M > 0. \tag{2}$$

82 *where*

$$|\mathcal{W}_{j^*,j}(t)| = |\{s < t, w(s) \in \mathcal{W}_{j^*,j}\}|.$$

83 *Proof.* According to the definitions in Appendix A.1, it holds that

$$\mathbb{P}\{\frac{\nu_{j^*}(t)}{\nu_j(t)} \leq M\} = \mathbb{P}\{S_t(j^*, j) \leq \log M\}.$$

84 For $\gamma \in [-1, 0]$, it holds that

$$\mathbb{P}\{S_t(j^*, j) \leq \log M\} = \mathbb{P}\{S_t(j^*, j) \leq 0\} + \mathbb{P}\{S_t(j^*, j) \in (0, \log M]\}$$

$$\leq \frac{\mathbb{E}[e^{\gamma S_t(j^*, j)}]}{\mathbb{E}[e^{\gamma S_t(j^*, j)}|S_t(j^*, j) \leq 0]} + \frac{\mathbb{E}[e^{\gamma S_t(j^*, j)}]}{\mathbb{E}[e^{\gamma S_t(j^*, j)}|S_t(j^*, j) \in (0, \log M]]}$$

$$\leq \mathbb{E}[e^{\gamma S_t(j^*, j)}] + \frac{\mathbb{E}[e^{\gamma S_t(j^*, j)}]}{e^{-\log M}} = \mathbb{E}[e^{\gamma S_t(j^*, j)}](1 + M).$$

85 We now consider $\tilde{H}_s(\gamma; j^*, j) = \mathbb{E}[(H_s(j^*, j))^\gamma]$. In case $w(s) \notin \mathcal{W}_{j^*, j}$, it holds that $H_s(j^*, j) = 1$
86 with probability 1, and therefore $\tilde{H}_s(\gamma; j^*, j) = 1$ for every $\gamma \in [-1, 0]$. On the other hand, consider
87 the case where $w(s) = w \in \mathcal{W}_{j^*, j}$. Then, for $\gamma = -1$, it holds that

$$\tilde{H}_s(-1; j^*, j) = q_{w,j^*}\left(\frac{\alpha_w}{1 - \alpha_w}\right)^{-1} + (1 - q_{w,j^*})\left(\frac{1 - \alpha_w}{\alpha_w}\right)^{-1}$$

88 which is lower or equal than 1 since $q_{w,j^*} \geq \alpha_w > 1/2$. Moreover, note that $\tilde{H}_s(0; j^*, j) = 1$ and
89 since $\tilde{H}_s(\cdot; j^*, j)$ is a convex function, it holds that $\tilde{H}_s(-1/2; j^*, j) < 1$. Hence, if $w(s) \in \mathcal{W}_{j^*, j}$,
90 it holds

$$\mathbb{E}[(H_s(j^*, j))^{-1/2}] = q_{w,j^*}\sqrt{\frac{1 - \alpha_w}{\alpha_w}} + (1 - q_{w,j^*})\sqrt{\frac{\alpha_w}{1 - \alpha_w}}$$

$$\leq \alpha_w\sqrt{\frac{1 - \alpha_w}{\alpha_w}} + (1 - \alpha_w)\sqrt{\frac{\alpha_w}{1 - \alpha_w}} = 2\sqrt{\alpha_w}\sqrt{1 - \alpha_w}.$$

91 Finally, observe that

$$2\sqrt{\alpha_w}\sqrt{1 - \alpha_w} = \sqrt{1 - \Delta_w^2} \leq \sqrt{1 - \Delta_m^2}$$

92 The proof is concluded by observing that

$$\frac{1}{1 + M}\mathbb{P}\left\{\frac{\nu_{j^*}(t)}{\nu_j(t)} \leq M\right\} \leq \mathbb{E}[e^{-S_t(j^*)/2}] = \prod_{s<t}\mathbb{E}[(H_s(j^*, j))^{-1/2}]$$

$$= \prod_{s<t}\tilde{H}_s(1/2; j^*, j) = \prod_{s<t, w(s) \in \mathcal{W}_{j^*, j}}\tilde{H}_s(1/2; j^*, j)$$

$$\leq (1 - \Delta_m^2)^{|\mathcal{W}_{j^*, j}(t)|/2}.$$

93 $\qquad\qquad\qquad\qquad\qquad\qquad\qquad\qquad\qquad\qquad\qquad\qquad\qquad\qquad\qquad\qquad\qquad\qquad\square$

94 The argument in the above proof is similar to [1, Lemma 1]. The important difference is that in [1]
95 every action is able to distinguish hypotheses $j^*$ and $j$, and therefore the exponent on the right-hand-
96 side of (2) is $t$ in that case, instead of $|\mathcal{W}_{j^*, j}(t)|$. Our model only satisfies Assumption 2; however if
97 a given action selection policy continues exploring each pairs of hypotheses, we deduce the following
98 corollary.

99 **Corollary 1.** *It holds that*

$$\lim_{t \to \infty} |\mathcal{W}_{j^*, j}(t)| = \infty \qquad \Rightarrow \qquad \lim_{t \to \infty}\mathbb{P}\left\{\frac{\nu_{j^*}(t)}{\nu_j(t)} < M\right\} = 0 \quad \forall M > 0.$$

100 As a consequence of Corollary 1, we deduce that any candidate policy for action selection shouldn't
101 have the property where there exists $\bar{t} > 0$ and $\bar{j} \in \mathcal{J}$ such that $w(s) \notin \mathcal{W}_{\bar{j}, j^*}$ for all $s \geq \bar{t}$. Indeed,
102 in such a case that policy would not be able to completely distinguish $\bar{j}$ from the true hypothesis $j^*$.

103 **A.4 Proof of Theorem 2**

104 Before the actual proof of Theorem 2 we need to show that, by using the IBAG algorithm, the decision
105 maker is expected to obtain a positive amount of information at each step.

106 **Proposition 5** (S). *Under the* IBAG*, there exists $K > 0$ such that*

$$\mathbb{E}[U_{t+1} - U_t] \geq K,$$

107 *where*

$$U_t = \log \frac{\nu_{j^*}(t)}{1 - \nu_{j^*}(t)}.$$

108 *Proof.* Denote by $w^D = w^D(t)$ the class of workers chosen by the IBAG algorithm at step $t$, and by
109 $\nu_{j^*} = \nu_{j^*}(t)$. Observe that

$$\mathbb{E}[U_{t+1} - U_t]$$

$$= \mathbb{E}[\log \frac{f(Y_t, j^*, w^D)}{\frac{\sum_{j \neq j^*} f(Y_t, j, w^D)\nu_j(t)}{\sum_{j \neq j^*} \nu_j(t)}}]$$

$$= q_{w^D, j^*} \log \frac{\alpha_{w^D}}{\alpha_{w^D} - \frac{\Delta_{w^D} \nu_{w^D, -j^*}}{1 - \nu_{j^*}}} + (1 - q_{w^D, j^*}) \log \frac{1 - \alpha_{w^D}}{1 - \alpha_{w^D} + \frac{\Delta_{w^D} \nu_{w^D, -j^*}}{1 - \nu_{j^*}}}$$

$$\geq \alpha_{w^D} \log \frac{\alpha_{w^D}}{\alpha_{w^D} - \frac{\Delta_{w^D} \nu_{w^D, -j^*}}{1 - \nu_{j^*}}} + (1 - \alpha_{w^D}) \log \frac{1 - \alpha_{w^D}}{1 - \alpha_{w^D} + \frac{\Delta_{w^D} \nu_{w^D, -j^*}}{1 - \nu_{j^*}}}$$

110

$$= D_{KL}\left(\left(\frac{1 + \Delta_{w^D}}{2}, \frac{1 - \Delta_{w^D}}{2}\right) \| \left(\frac{1 + \Delta_{w^D}}{2} - \frac{\Delta_{w^D} \nu_{w^D, -j^*}}{1 - \nu_{j^*}}, \frac{1 - \Delta_{w^D}}{2} + \frac{\Delta_{w^D} \nu_{w^D, -j^*}}{1 - \nu_{j^*}}\right)\right)$$

$$\geq D_{KL}\left(\left(\frac{1 + \Delta^m}{2}, \frac{1 - \Delta^m}{2}\right) \| \left(\frac{1 + \Delta^m}{2} - \Delta^m \frac{\nu_{w^D, -j^*}}{1 - \nu_{j^*}}, \frac{1 - \Delta^m}{2} + \Delta^m \frac{\nu_{w^D, -j^*}}{1 - \nu_{j^*}}\right)\right)$$

$$=: K(\nu_{w^D, -j^*}, \nu_{j^*}) \geq 0$$

111 where $D_{KL}(\cdot \| \cdot)$ denotes the Kullback-Leibler divergence. Observe that

$$\frac{\partial K(x, y)}{\partial x} \geq 0, \qquad \frac{\partial K(x, y)}{\partial y} \geq 0,$$

112 and we aim to bound $K(x, y)$ away from zero with high probability. We fix $\tilde{\nu} < 1/J$ and distinguish
113 two possible cases:

114

115 *Case 1:* The decision maker is in an explorative phase, i.e., it does not have a clear feeling about
116 which type the incoming job is. In this phase, there exists $\bar{w} \in \mathcal{W}$ such that

$$\nu_{-\bar{w}} \geq \tilde{\nu}.$$

117 This yields that

$$\nu_{j^*} \geq 0, \qquad \nu_{w^D, -j^*} \geq \nu_{-w^D} \geq \tilde{b}\nu_{-\bar{w}} \geq \tilde{b}\tilde{\nu},$$

118 where $\tilde{b}$ is defined in Lemma 3(S) which is proved in Appendix B. Therefore

$$K(\nu_{w^D, -j^*}, \nu_{j^*}) \geq K(\tilde{b}\tilde{\nu}, 0) > 0.$$

119 *Case 2:* The decision maker is in an exploitative phase, i.e., it does have a clear feeling about which
120 type the incoming job is. In this phase, for every $w \in \mathcal{W}$, it holds that

$$\nu_{-w} < \tilde{\nu}.$$

121 The following lemma states that indeed in this case there is a job-type which is clearly the most likely
122 type of the incoming job. The proof is provided in Appendix B.

123 **Lemma 2** (S). *If $\nu_{-w} < \frac{1}{J}$ for every $w \in \mathcal{W}$, then there exists $\bar{j} \in \mathcal{J}$ such that*

$$\bigcap \mathcal{J}_{+w} = \{\bar{j}\}, \qquad \mathcal{J}_{+w} = \begin{cases} \mathcal{J}_w, & \text{if } \sum_{j \in \mathcal{J}_w} \nu_j \geq \sum_{j \notin \mathcal{J}_w} \nu_j, \\ \mathcal{J} \setminus \mathcal{J}_w, & \text{if } \sum_{j \in \mathcal{J}_w} \nu_j < \sum_{j \notin \mathcal{J}_w} \nu_j. \end{cases}$$

124 At this point, we distinguish two subcases:

2a) Assume $\bar{j} = j^*$. It means that we are on the correct path towards the end of the learning
process, and

$$\nu_{w^D, -j^*} = \nu_{-w^D}, \qquad \nu_{j^*} \geq 1 - \frac{J-1}{\tilde{b}} \nu_{-w^D},$$

where the second relation holds due to Lemma 3(S) and

$$1 = \nu_{j^*} + \sum_{j \neq j^*} \nu_j \leq \nu_{j^*} + \sum_{j \neq j^*} \nu_{-w(j)} \leq \nu_{j^*} + (J-1)\frac{\nu_{-w^D}}{\tilde{b}}.$$

Hence,

$$K(\nu_{w^D, -j^*}, \nu_{j^*}(t)) \geq K(\nu_{-w^D}, 1 - \frac{J-1}{\tilde{b}} \nu_{-w^D}) = K(1, 1 - \frac{J-1}{\tilde{b}}) > 0.$$

2b) Assume $\bar{j} \neq j^*$. It means that we are on the wrong path towards the end of the learning
process, and we would like to show that this is unlikely to happen. Denote by $w(j)$ a class
of workers belonging to $\mathcal{W}_{j,\bar{j}}$, it holds that

$$\nu_{j^*}(t) < \nu_{-w(j^*)} \leq \tilde{\nu}, \qquad \nu_{\bar{j}}(t) \geq 1 - (J-1)\tilde{\nu},$$

since

$$1 = \nu_{\bar{j}} + \sum_{j \neq \bar{j}} \nu_j \leq \nu_{\bar{j}} + \sum_{j \neq \bar{j}} \nu_{-w(j)} \leq \nu_{\bar{j}} + (J-1)\tilde{\nu}.$$

Therefore

$$\frac{\nu_{j^*}(t)}{\nu_{\bar{j}}(t)} \leq \frac{\tilde{\nu}}{1 - (J-1)\tilde{\nu}}$$

and Proposition 1 yields that

$$\mathbb{P}\{\frac{\nu_{j^*}(t)}{\nu_{\bar{j}}(t)} \leq \frac{\tilde{\nu}}{1 - (J-1)\tilde{\nu}}\} \leq \frac{\tilde{\nu}}{1 - (J-1)\tilde{\nu}}$$

135 Hence in this second phase, we obtain that

$$\begin{aligned}
\mathbb{E}[U_{t+1} - U_t] &= \mathbb{E}[U_{t+1} - U_t | \bar{j} = j^*]\mathbb{P}\{\bar{j} = j^*\} + \mathbb{E}[U_{t+1} - U_t | \bar{j} \neq j^*]\mathbb{P}\{\bar{j} \neq j^*\} \\
&\geq \mathbb{E}[U_{t+1} - U_t | \bar{j} = j^*]\mathbb{P}\{\bar{j} = j^*\} \\
&\geq \mathbb{E}[U_{t+1} - U_t | \bar{j} = j^*](1 - \mathbb{P}\{\frac{\nu_{j^*}(t)}{\nu_{\bar{j}}(t)} \leq \frac{\tilde{\nu}}{1 - (J-1)\tilde{\nu}}\}) \\
&\geq K(1, 1 - \frac{J-1}{\tilde{b}})\frac{1 - J\tilde{\nu}}{1 - J\tilde{\nu} + \tilde{\nu}}.
\end{aligned}$$

136 Define the following events

$$A_{\tilde{\nu}}(t) = \{\exists \bar{w} : \nu_{-\bar{w}}(t) \geq \tilde{\nu}\}, \qquad B_{\tilde{\nu}}(t) = A_{\tilde{\nu}}(t)^C.$$

137 We just showed that

$$\mathbb{E}[U_{t+1} - U_t | A_{\tilde{\nu}}(t)] \geq K(\tilde{b}\tilde{\nu}, 0) =: K_A(\tilde{\nu})$$

138 and that

$$\mathbb{E}[U_{t+1} - U_t | B_{\tilde{\nu}}(t)] \geq K(1, 1 - \frac{J-1}{\tilde{b}})\frac{1 - J\tilde{\nu}}{1 - J\tilde{\nu} + \tilde{\nu}} =: K_B(\tilde{\nu}).$$

139 Observing that

$$\begin{aligned}
\mathbb{E}[U_{t+1} - U_t] &= \mathbb{E}[U_{t+1} - U_t | A_{\tilde{\nu}}(t)]\mathbb{P}(A_{\tilde{\nu}}(t)) + \mathbb{E}[U_{t+1} - U_t | B_{\tilde{\nu}}(t)]\mathbb{P}(B_{\tilde{\nu}}(t)) \\
&\geq K_A(\tilde{\nu})\mathbb{P}(A_{\tilde{\nu}}(t)) + K_B(\tilde{\nu})\mathbb{P}(B_{\tilde{\nu}}(t)) \\
&\geq \min\{K_A(\tilde{\nu}), K_B(\tilde{\nu})\}.
\end{aligned}$$

140 Define

$$K_{\tilde{\nu}} = \min\{K_A(\tilde{\nu}), K_B(\tilde{\nu})\} > 0,$$

141    so as that, for $\tilde{\nu} = \frac{1}{J^2}$, it holds that

$$K_{\frac{1}{J^2}} = \min\{K(\frac{\tilde{b}}{J^2}, 0), K(1, 1 - \frac{J-1}{\tilde{b}})\frac{J^2 - J}{J^2 - J + 1}\}.$$

142    Note that as $J$ grows large both $K(\frac{\tilde{b}}{J^2}, 0)$ and $K(1, 1 - \frac{J-1}{\tilde{b}})$ converges to zero as $\log(\frac{J}{1+J})$.    $\square$

143    *Proof.* (of Theorem 2.) Observe that by the definition of $T^*$, it holds that $T^* \leq T(j^*)$ where

$$T(j^*) = \inf_t \{\nu_{j^*}(t) > 1 - \delta | j^*\}.$$

144    Hence,

$$\mathbb{E}[T^*] \leq \mathbb{E}[T(j^*)] = \sum_t \mathbb{P}\{T(j^*) > t\}.$$

145    Observe that

$$\nu_{j^*} \leq 1 - \delta \qquad \Rightarrow \qquad \frac{\nu_{j^*}}{1 - \nu_{j^*}} \leq \frac{1 - \delta}{\delta} = \frac{1}{\delta} - 1 < \frac{1}{\delta},$$

146    and therefore

$$\{T(j^*) > t\} \Rightarrow B(t), \qquad B(t) = \{U_t \leq -\log \delta\},$$

147    where

$$U_t = \log \frac{\nu_{j^*}(t)}{1 - \nu_{j^*}(t)}.$$

148    Note that

$$\mathbb{P}\{B(t)\} = \mathbb{P}\{U_t \leq -\log \delta\} = \mathbb{P}\{U_t - \mathbb{E}[U_t] \leq -\log \delta - \mathbb{E}[U_t]\},$$

149    and from Proposition 5(S) we obtain

$$\mathbb{P}\{B(t)\} \leq \mathbb{P}\{U_t - \mathbb{E}[U_t] \leq -\log \delta - U_0 - tK\}.$$

150    At this point, consider any $t \geq \bar{t}(\delta) := \frac{2}{K}\eta(\delta)$, with $K = K_{\frac{1}{J^2}}$ and $\eta(\delta) = -\log \delta - U_0 > 0$ then

$$\mathbb{P}\{B(t)\} \leq \mathbb{P}\{-U_t + \mathbb{E}[U_t] \geq tK - \eta(\delta)\}$$
$$\leq \mathbb{P}\{|U_t - \mathbb{E}[U_t]| \geq tK - \eta(\delta)\}$$
$$\leq e^{-\frac{(tK - \eta(\delta))^2}{2Ht}}$$

151    where

$$H = 2\frac{1 + \Delta^M}{1 - \Delta^M} \geq \max_{s < t} |U_s - U_{s-1} - \mathbb{E}[U_s - U_{s-1}]|.$$

152    and the last inequality is due to Azuma's inequality.

153    Hence,

$$\mathbb{E}[T(j^*)] \leq \bar{t}(\delta) + \sum_{t \geq \bar{t}} \mathbb{P}\{T(j^*) > t\}$$
$$\leq \bar{t}(\delta) + \sum_{t \geq \bar{t}(\delta)} e^{-\frac{(tK - \eta(\delta))^2}{2Ht}}$$
$$= \bar{t}(\delta) + \sum_{t \geq \bar{t}(\delta)} e^{-\frac{tK^2}{2H}} e^{\frac{K\eta(\delta)}{H}} e^{\frac{-\eta(\delta)^2}{2Ht}}$$
$$\leq \bar{t}(\delta) + e^{\frac{K\eta(\delta)}{H}} \sum_{t \geq \bar{t}(\delta)} e^{-\frac{tK^2}{2H}}$$
$$= \bar{t}(\delta) + e^{\frac{K\eta(\delta)}{H}} \frac{e^{-\frac{\bar{t}(\delta)K^2}{2H}}}{1 - e^{-\frac{K^2}{2H}}} = \bar{t}(\delta) + \frac{1}{1 - e^{-\frac{K^2}{2H}}},$$

154 Theorem 2 follows by defining

$$K_1^u = \frac{2}{K}, \qquad K_0^u = -\frac{2U_0}{K} + \frac{1}{1 - e^{-\frac{K^2}{2H}}}.$$

155 For the sake of completeness, note that $H$ is constant in $J$, and

$$U_0 = -\log(J-1), \qquad \frac{1}{K} \sim \frac{1}{\log(\frac{J}{1+J})}.$$

156 $\qquad\qquad\qquad\qquad\qquad\qquad\qquad\qquad\qquad\qquad\qquad\qquad\qquad\qquad\qquad\qquad\qquad$ $\square$

## 157 B  Proof of the technical lemmas

158 *Proof of Lemma 1.* Given that at time $t$ we observe the belief vector $\boldsymbol{\nu} = \boldsymbol{\nu}(t) \in \mathbb{P}(\mathcal{J})$, and a response
159 is asked to a worker in class $w = w(t)$. For every worker $w \in \mathcal{W}$ and $j \in \mathcal{J}$, define

$$\nu_{w,+j} = \sum_{j \in \mathcal{J}_{w,+j}} \nu_j, \qquad \nu_{w,-j} = \sum_{j \in \mathcal{J}_{w,-j}} \nu_j.$$

160 Then if $y = g_{w,j^*}$, i.e., a correct response is observed,

$$j \in \mathcal{J}_{w,+j^*} \quad \Rightarrow \quad \nu_j(t+1) = \frac{\nu_j(1+\Delta_w)}{\nu_{w,+j^*}(1+\Delta_w) + \nu_{w,-j^*}(1-\Delta_w)} \geq \nu_j,$$

$$j \notin \mathcal{J}_{w,+j^*} \quad \Rightarrow \quad \nu_j(t+1) = \frac{\nu_j(1-\Delta_w)}{\nu_{w,+j^*}(1+\Delta_w) + \nu_{w,-j^*}(1-\Delta_w)} \leq \nu_j,$$

161 while, on the other hand, if $y = -g_{w,j^*}$, i.e., a wrong response is observed,

$$j \in \mathcal{J}_{w,+j^*} \quad \Rightarrow \quad \nu_j(t+1) = \frac{\nu_j(1-\Delta_w)}{\nu_{w,+j^*}(1-\Delta_w) + \nu_{w,-j^*}(1+\Delta_w)} \leq \nu_j,$$

$$j \notin \mathcal{J}_{w,+j^*} \quad \Rightarrow \quad \nu_j(t+1) = \frac{\nu_j(1+\Delta_w)}{\nu_{w,+j^*}(1-\Delta_w) + \nu_{w,-j^*}(1+\Delta_w)} \geq \nu_j.$$

162 The analysis of $\nu_{j^*}(t+1)/\nu_j(t+1)$ in the various cases conclude the proof.
163

164 *Proof of Lemma 4.* Lemma 4 is a consequence of the following stronger result.
165 **Lemma 3** (S). *Consider $x, y \in [0, \frac{1}{2}]$ such that $G(x, \Delta^M) \geq G(y, \Delta^m)$, then*

$$x \geq \tilde{b}y, \qquad \tilde{b} := \frac{\Delta^m}{\Delta^M}. \tag{3}$$

166 In fact, from the monotonicity properties of the function $G(\cdot, \cdot)$, at time $t$ it holds that

$$G(\nu_{-w^D(t)}, \Delta^M) \geq G(\nu_{-w^D(t)}, \Delta_{w^D(t)}) \geq G(\nu_{-w}, \Delta_w) \geq G(\nu_{-w}, \Delta^m),$$

167 and Lemma 3(S) yields $\nu_{-w^D(t)} \geq \tilde{b}\nu_{-w}$ for every $w \in \mathcal{W}$.

168 It remains to prove Lemma 3(S) whose proof is done via a contradiction argument. Assume that
169 $x < \tilde{b}y$. In this case, it holds that

$$\begin{aligned}
G(x, \Delta^M) &< G(\tilde{b}y, \Delta^M) \\
&= \frac{(\Delta^M)^2 (\tilde{b}y)^2}{1 - (\Delta^M)^2 (1 - 2\tilde{b}y)^2} \\
&= \frac{(\Delta^m)^2 y^2}{1 - (\Delta^M)^2 (1 - 2\tilde{b}y)^2} \\
&= \frac{(\Delta^m)^2 y^2}{1 - (\Delta^m)^2 (1 - 2y)^2} = G(y, \Delta^m),
\end{aligned}$$

170 which is a contradiction. Note that the last equality follows since

$$\frac{1}{1 - (\Delta^M)^2(1 - 2\tilde{b}y)^2} \leq \frac{1}{1 - (\Delta^m)^2(1 - 2y)^2}$$

$$\Longleftrightarrow \quad (\Delta^M)^2(1 - 2\tilde{b}y)^2 \leq (\Delta^m)^2(1 - 2y)^2$$

$$\Longleftrightarrow \quad \Delta^M(1 - 2\tilde{b}y) \leq \Delta^m(1 - 2y)$$

$$\Longleftrightarrow \quad y(\Delta^m - \tilde{b}\Delta^M) \leq \frac{1}{2}(\Delta^m - \Delta^M)$$

$$\Longleftrightarrow \quad y \leq \frac{1}{2}\frac{(\Delta^m - \Delta^M)}{(\Delta^m - \tilde{b}\Delta^M)}$$

171 which is true since $y \leq 1/2$ and

$$\frac{(\Delta^m - \Delta^M)}{(\Delta^m - \tilde{b}\Delta^M)} \geq 1.$$

172 *Proof of Lemma 2(S).* First of all, observe that $|\bigcap \mathcal{J}_{+w}(t)| \leq 1$. In fact, assume $\{j_1, j_2\} \in \bigcap \mathcal{J}_{+w}(t)$,
173 then consider $\bar{w} \in \mathcal{W}_{j_1,j_2}$, it yields a a contradiction since it is not possible that

$$\{j_1, j_2\} \in \mathcal{J}_{+\bar{w}}(t).$$

174 Now, assume that $\bigcap \mathcal{J}_{+w}(t) = \emptyset$, then, for every $j \in \mathcal{J}$ it is possible to identify $w(j) \in \mathcal{W}$ such that
175 $j \in \mathcal{J}_{-w}(t)$. For this reason, it holds that

$$1 = \sum_{j \in \mathcal{J}} \nu_j(t) \leq \sum_{j \in \mathcal{J}} \nu_{-w(j)}(t) < J\frac{1}{J} = 1.$$

176 Hence, there exists $\bar{j} \in \bigcap \mathcal{J}_{+w}(t)$.
177

# C  On the effect of the slack

179 In this section we investigate how different the choices of the IBAG algorithm would be if, instead
180 of $\alpha_w$, we had at our disposal the exact skill parameters $q_{w,j}$. So as to gain better insights, in this
181 section we assume $q_{w,j}$ to be independent from $j$, i.e.,

$$q_{w,j} = q_w \in [\alpha_w, \alpha_w + \sigma_w],$$

182 and we aim to capture the effect of $\sigma_w$ on the algorithm decision. Define

$$\Delta_{q(w)} = 2q_w - 1 \in [\Delta_w, \bar{\Delta}_w], \qquad \Delta_w = 2\alpha_w - 1, \quad \bar{\Delta}_w = \Delta_w + 2\sigma_w.$$

183 With the knowledge of $q_w$, the Incomplete Bayesian updating rule described in Section 3 coincides
184 exactly with the classical Bayesian updating rule presented in Section 3 as well. In particular

$$\mathbb{E}[\nu_{j^*}(t+1)|\boldsymbol{\nu}, w(t) = w] - \nu_{j^*}(t) = 4\nu_{j^*}(t)\frac{\Delta_{q(w)}^2\nu_{w,-j^*}(t)^2}{1 - \Delta_{q(w)}^2(1 - \nu_{w,-j^*}(t))^2}$$

185 and denote by $w^B(t)$ the class of workers picked by IBAG in this case, i.e.,

$$w^B(t) = \arg\max_{w \in \mathcal{W}}\{G(\nu_{-w}(t), \Delta_{q(w)})\}, \qquad G(v, d) = \frac{d^2v^2}{1 - d^2(1 - 2v)^2}.$$

186 We recall that the IBAG algorithm, which only knows $\alpha_w$, at time $t$ picks the class of workers $w^D(t)$
187 maximizing

$$w^D(t) = \arg\max_{w \in \mathcal{W}}\{G(\nu_{-w}(t), \Delta_w)\},$$

188 and we would like to better understand under which condition the choices made in the two cases are
189 different, i.e,

$$w^B(t) \neq w^D(t),$$

190 and, in case they are different, what is the impact of the error on the performance, i.e.,

$$\text{Err}(\boldsymbol{\nu}) = \mathbb{E}[\nu_{j^*}(t+1)|\boldsymbol{\nu} = \boldsymbol{\nu}(t), w(t) = w^B(t)] - \mathbb{E}[\nu_{j^*}(t+1)|\boldsymbol{\nu} = \boldsymbol{\nu}(t), w(t) = w^D(t)].$$

191 Note that, it follows from the definition of $w^B(t)$ that $\text{Err}(\boldsymbol{\nu}) \geq 0$.

192   **Lemma 4** (S). *If for every $w \neq w^D(t)$ it holds that*

$$G(\nu_{-w^D(t)}(t), \Delta_{w^D(t)}) \geq G(\nu_{-w}(t), \bar{\Delta}_w)$$

193   *then $w^D(t) = w^B(t)$.*

194   *Proof.* Recall that $G(v, d)$ is increasing in $d$, hence

$$G(\nu_{-w^D(t)}(t), \Delta_{q(w^D(t))}) \geq G(\nu_{-w^D(t)}(t), \Delta_{w^D(t)}),$$

195   and

$$G(\nu_{-w}(t), \bar{\Delta}_w) \geq G(\nu_{-w}(t), \Delta_{q(w)}), \qquad \forall w \in \mathcal{W}.$$

196   Hence, for every $w \neq w^D(t)$, the hypothesis of the lemma holds. $\qquad\square$

197   In particular, this lemma states that if a class of worker is *much more convenient* than the others when
198   only the lower bound $\alpha_w$ are known, then the same holds when the probabilities $q_w$ are known. Here
199   we investigate what kind of threshold determines a worker to be much more convenient than the
200   others. For every $w \in \mathcal{W}$ define the following quantities

$$r_w = G(\nu_{-w^D(t)}(t), \Delta_{w^D(t)}) - G(\nu_{-w}(t), \Delta_w),$$
$$s_w = G(\nu_{-w}(t), \bar{\Delta}_w) - G(\nu_{-w}(t), \Delta_w).$$

201   Note that $r_w$ represents by how much $w^D(t)$ is more convenient than $w$ given that only $\alpha_w$ is known,
202   in particular

$$G(\nu_{-w^D(t)}(t), \Delta_{w^D(t)}) \geq G(\nu_{-w}(t), \bar{\Delta}_w) \quad \Longleftrightarrow \quad r_w \geq s_w.$$

203   **Proposition 6** (S). *It holds that*

$$\lim_{\sigma_w \to 0} s_w = 0, \qquad \lim_{\nu_{-w}(t) \to 0} s_w = 0.$$

204   *Proof.* Note that

$$s_w = G(\nu_{-w}(t), \bar{\Delta}_w) - G(\nu_{-w}(t), \Delta_w)$$
$$= 4 \frac{\nu_{-w}(t)^2 \sigma_w (\Delta_w + \sigma_w)}{\left(1 - (\Delta_w + 2\sigma_w)^2 (1 - 2\nu_{-w}(t))^2\right)\left(1 - \Delta_w^2 (1 - 2\nu_{-w}(t))^2\right)}$$
$$\leq 4 \frac{\nu_{-w}(t)^2 \sigma_w (\Delta_w + \sigma_w)}{\left(1 - (2\sigma_w + \Delta_w)^2 (1 - 2\nu_{-w}(t))^2\right)^2}.$$

205   The proposition follows by taking the limits. $\qquad\square$

206   As a consequence of this proposition, it follows that for $\sigma_w$ or $\nu_{-w}(t)$ sufficiently low, it holds that
207   $r_w \geq s_w$. Recall that when this happens for each $w \neq w^D(t)$, it follows that $w^D(t) = w^B(t)$.

208   Nevertheless, even when the worker chosen is not the optimal one, the error incurred is not large in
209   many circumstances. The following proposition bounds $\mathrm{Err}(\boldsymbol{\nu})$ with a linear function of $\max_w\{s_w\}$,
210   whose dependence on $\sigma_w$ has been pointed out in the proof of Proposition 6(S).

211   **Proposition 7** (S). *It holds that*

$$Err(\boldsymbol{\nu}) \leq 4\nu_{j^*}(t)s_{w^B(t)} \leq 4\nu_{j^*}(t)\max_w\{s_w\}.$$

212   *Proof.* Observe that due to the definition of $G$ and $\mathrm{Err}(\boldsymbol{\nu})$, we only need to to upper bound the
213   following difference

$$G(\nu_{-w}(t), \Delta_{q(w)}) - G(\nu_{-w^D}(t), \Delta_{q(w^D)})$$

214   under the constraint that

$$G(\nu_{-w}(t), \Delta_w) \leq G(\nu_{-w^D}(t), \Delta_{w^D}).$$

215   Observe that

$$G(\nu_{-w}(t), \Delta_{q(w)}) - G(\nu_{-w^D}(t), \Delta_{q(w^D)})$$
$$\leq G(\nu_{-w}(t), \bar{\Delta}_w) - G(\nu_{-w^D}(t), \Delta_w)$$
$$\leq G(\nu_{-w}(t), \bar{\Delta}_w) - G(\nu_{-w}(t), \Delta_w) = s_w.$$

216 $\qquad\square$