[Reviews · NeurIPS 2017]

Reviewer 1



The authors presented a new algorithm, named the IBAG algorithm, for active sequential learning. The main idea is that it requires only a limited knowledge of a set of target actions. The authors did an extensive theoretical formulation and evaluation of their approach, and showed that it outperforms, in most cases, the benchmark approach (Chernoff’s algorithm). Overall, I think that this paper adds value to existing literature on hypotheses testing under the active learning setting and in presence of limited information.

Reviewer 2



The paper studies the problem of active hypothesis testing with limited information. More specifically, denoting the probability of ‘correct indication’ by q(j,w) if the true hypothesis is j and the chosen action is w, they assume that q(j,w) is bounded below by \alpha(w) for all j, and this quantity is available to the controller. They first study the incomplete-bayesian update rule, where all q’s are replaced with alpha’s, hence approximate, and show that with this IB update rule, one needs at least O(log(1/\delta)) to achieve (1-\delta) posterior error probability. Then, they show that their gradient-based policy achieves the order-wise optimal sample complexity. I have a few technical questions. On the upper bound: The authors claim that the upper bound is obtained by considering the worst case scenario where q(w,j) = \alpha(w) for every w if j is the true hypothesis. If that’s the case, how is different the analysis given in this paper from that for Bayesian update case? (If my understanding is correct, q(w,j) = \alpha(w) implies IB update rule = Bayesian update rule.) Further, what can happen in reality is that q(w,j) is arbitrarily close to 1, and alpha(w) is arbitrarily close to 1/2. In this case, can the upper bound provide any guarantee? On the lower bound: Can this bound recover the existing lower bounds for the case of complete information? On simulation results: It will be great if the authors can present the lower & upper bounds on E[T] and compare them with the simulation results.

Reviewer 3



This paper considers active hypothesis testing where the algorithm only receives an indirect feedback. It proposes a method based on incomplete Bayesian update, and at each step greedily chooses the action that maximizes a lower bound on the expected posterior of the true hypothesis. It gives matching lower bound and upper bound on the number of samples required w.r.t. delta, the confidence parameter. This paper is clearly written, and I believe its results are correct (I didn't read the appendix). It considers a new model, and the algorithm proposed is new (but similar to the existing methods in [11] and [13]) My main concern is the motivation of this paper. This paper proposes a new model of active hypothesis testing, but gives little justification for it. The paper does provide an example in section 5 (I think it would be better to move it into section 1 or 2), but even that example is unnatural to me: why don't we just let the workers output the labels? How can you know the quality of the worker? Another issue is that the analysis is limited: 1. The upper and lower bounds are only with respect to delta. What is the dependency on |J|, |W|, or some other quantities? 2. How is the sample complexity of the proposed method compared with other methods like the Chernoff algorithm, GBS algorithm, and a naive passive method? The paper provides numerical results, but I would expect to see theoretical comparisons.